# Design and Evaluation of an Augmented Reality-Based Exergame System to Reduce Fall Risk in the Elderly

**DOI:** 10.3390/ijerph17197208

**Published:** 2020-10-01

**Authors:** Meiling Chen, Qingfeng Tang, Shoujiang Xu, Pengfei Leng, Zhigeng Pan

**Affiliations:** 1School of Health Management, Hangzhou Normal University, Hangzhou 310036, China; chenmeiling920@zcmu.edu.cn (M.C.); tqf1013@stu.hznu.edu.cn (Q.T.); shoujiang.xu@jsfpc.edu.cn (S.X.); lengpengfei@stu.hznu.edu.cn (P.L.); 2School of Humanity and Management, Zhejiang Chinese Medical University, Hangzhou 310053, China; 3The University Key Laboratory of Intelligent Perception and Computing of Anhui Province, Anqing Normal University, Anqing 246011, China; 4Institute of VR and Intelligent System, Hangzhou Normal University, Hangzhou 310036, China

**Keywords:** fall risk, augmented reality, exergame, cognitive–motor intervention, UEQ-S, user experience

## Abstract

Falls are a major public health concern in today’s aging society. Virtual reality (VR) technology is a promising method for reducing fall risk. However, the absence of representations of the user’s body in a VR environment lessens the spatial sense of presence. In terms of user experience, augmented reality (AR) can provide a higher degree of presence and embodiment than VR. We developed an AR-based exergame system that is specifically designed for the elderly to reduce fall risk. Kinect2.0 was used to capture and generate 3D models of the elderly and immerse them in an interactive virtual environment. The software included three functional modules: fall risk assessment, cognitive–motor intervention (CMI) training, and training feedback. The User Experience Questionnaire (UEQ-S) was used to evaluate user experience. Twenty-five elders were enrolled in the study. It was shown that the average scores for each aspect were: pragmatic quality score (1.652 ± 0.868); hedonic quality score (1.880 ± 0.962); and overall score was 1.776 ± 0.819. The overall score was higher than 0.8, which means that the system exhibited a positive user experience. After comparing the average score in a dataset product of UEQ-S Data Analysis Tool, it was found that the pragmatic quality aspect was categorized as good, while the hedonic quality aspect was categorized as excellent. It revealed a positive evaluation from users.

## 1. Introduction

In today’s aging society, falls are a major public health concern. According to the World Health Organization (WHO), approximately 28–35% of the elderly who are aged ≥65 years, experience falls yearly. The frequency of falls simultaneously increases with an increase in age and frailty level [1]. In China, falls are the leading cause of accidental or unintentional injurious deaths among the elderly who are aged ≥65 years. In old people who may be suffering from functional impairment, falls lead to injuries, fractures, disabilities, a lower quality of life, increased mortalities, and an economic burden [2].

Falls occur as a result of a complex interaction of risk factors [3]. The associated risk factors include advanced age [4], muscle weakness [5,6], gait disorders [7], balance impairment [8], cognitive deficits [9], and others. In terms of intervention effect or time and energy consumption, active exercise is the best way to prevent falls among the elderly [10]. It is recommended by the World Health Organization (WHO) that elderly people (≥65 years) should be involved in an aerobic physical activity for at least 150 min (moderate-intensity) or 75 min (high intensity) per week. Exercises such as body stretch, strength, body-balance, tai chi, and treadmill training can reduce the risk of falls among the elderly. Cognitive impairments, especially the decline in attention, memory, and executive function have been associated with falls among the elderly [9,11]. Posture and gait in humans are controlled by the cerebral cortex, spinal cord, and other motor centers. In addition, cognitive and sensory processing are involved, resulting in interactions between perceptual, cognitive, and motor systems. The elderly are more likely to fall when they are performing cognitive motor tasks during their daily activities. Therefore, cognitive–motor interference (CMI) is popularly used to improve gait and balancing functions in sports and during rehabilitation sessions [12,13].

Virtual reality (VR) is a promising solution for CMI. It combines entertainment, physical activities, and cognitive tasks to improve the consumability, safety, and compliance of the intervention. Studies have utilized commercial VR-based exergames that target a variety of physical functions, including balance and strength in the elderly. As a result, the feasibility and effectiveness of the VR technology in reducing fall risk among the elderly have been preliminarily confirmed [14,15,16]. Exergames based on VR technology can achieve the effect of traditional exercise training or better effects, as well as increasing the fun and safety of the whole rehabilitation process. A multicentric randomized controlled trial found that a combination of treadmill training and non-immersive virtual reality (VR) interventions aimed at improving body movements and cognition were less likely to cause falls compared to treadmill training alone [17].

A good user experience promotes a high quality interaction between the users and the system [18]. However, the elderly’s user experience on exergames was often ignored in previous studies. Most exergames were designed to entertain young people and not for balance rehabilitation purposes among the elderly. This inhibits the independent use and successful implementation of interventions that are based on exergames [16,19,20,21]. The absence of representations of the users’ body in the VR environment may lessen the sense of spatial presence [22]. In terms of user experience, augmented reality (AR) can provide a higher degree of presence and embodiment than VR, despite exhibiting advantages similar to VR in limb function recovery [23,24]. It combines real and virtual objects and runs interactively in a common real-time environment. This property is more acceptable and suitable for the elderly.

Because of the above, we developed an AR-based exergame system to reduce fall risk in the elderly and evaluated its user experience. This paper is structured as follows. Section 2 describes the Hardware Setup as well as the software exercise modules. Section 3 and Section 4 analyze the usability and user experience through questionnaires. Section 5 discusses the results, while Section 6 presents our conclusions.

## 2. System Design

### 2.1. The Hardware Setup

The hardware was composed of a computer, an LCD, audio and Kinect 2.0 sensors, which are respectively used in data processing, virtual scene display, sound feedback and motion interaction. The system used the bone tracking function of Kinect 2.0 to track the movement of the elderly in real-time. Kinect 2.0 was also used as an interactive device to perform cognition and movement training. The architecture of the comprehensive intervention system for preventing falls in the elderly is shown in Figure 1.

Kinect 2.0 is a motion-sensing interactive device designed by Microsoft. It consists of an infra-red (IR) projector, an IR camera, an RGB camera, and a multi-array microphone. It is important for acquiring real-time video, depth, and bone data. It also realizes voice input and recognition, therefore, discerning the goal of “the body is the controller”. When the Kinect’s skeletal data stream is powered on, Kinect can quickly detect a human body and track its bones in real-time when an elderly person is present in its field of vision. The Kinect 2.0 provides a 25-joint skeletal model that compares it with the 20-joint model given by the Kinect 1.0 (shown in Figure 2). Therefore, users do not have to wear sensors because interactions with the computer can be achieved through various gestures and postures. Kinect simplifies interactions and allows for more natural mappings, which eases entry into play if institutionalized elders lack gaming experience [21].

### 2.2. The Software Design

The software included the following functional modules: fall risk assessment, CMI training, and training feedback.

#### 2.2.1. Fall Risk Assessment

The fall risk assessment module utilized the Self-Rated Fall Risk Questionnaire [25], to identify elderly fall risk. The checklist was developed by the Greater Los Angeles VA Geriatric Research Education Clinical Center and affiliates. It was then validated as a fall risk self-assessment tool. There were 12 items in the checklist, which were scored by the two-point scoring method; “Yes” was the corresponding score while “no” was marked as 0. Item 1 “I have fallen in the past year” and item 2 “I use, or I have been advised to use a cane or walker to get around safely” were assigned 2 points. All the other entries were assigned 1 point. When the total score was ≥4 points, it indicated a high fall risk. Therefore, the higher the score, the greater the risk of falling. The assessment results were used as the main basis for training difficulty setting.

#### 2.2.2. CMI Training

Exercise (grade B) and selectively administering multifactorial interventions (grade C) are recommended by the USPSTF to prevent falls in community-based adults who are aged ≥65 years [10]. The most common exercise components were gait, balance, and functional training. Cognitive training includes attention training, executive function (EF) training, memory training, directional ability training and visuospatial ability training. Executive function (EF) and attention play important roles in gait and mobility [26]. Therefore, the CMI training module consisted of three exergames: Exercise 1, referred to as the “wall dodging game”, was a strengthening exercise for muscles; Exercise 2, referred to as the “fruits picking game” was a balance training exercise; Exercise 3, referred to as the “rats stomping game” is a gait training exercise. Explicit cognitive training tasks for attention, memory, and executive functions were added to each game as shown in Table 1.

The wall dodging game: to step into the VR world of the “Wall dodging” game, the participants had to fit through the hole in each wall as it came their way. Each wall presented a unique challenge involving quick thinking and motor dexterity. The exercise increased muscular strength, balance ability, attention and executive functions as the participants ducked, dipped, and dodged in order to avoid each of the walls. Some hole shapes in the wall are shown in Figure 3. They were designed to enhance lower limb strength and balance by exercises such as side-leg lift, squat, one-leg standing, lunge, lateral bending, and side lateral raise (Figure 3).The fruits picking game: The aim of this exercise was balance training. Three fruits were randomly displayed on the screen, and the participant was supposed to remember them in five seconds, and then try to catch a specified fruit by moving the body from side to side. Only by catching the specified fruit will a score be taken. Participants were penalized if they caught the wrong fruit or missed a catch. The movement of the body from side to side helped improve the body’s dynamic balance ability. To complete the game, they had to memorize three randomly occurring fruits in five seconds, which is designed to improve memory, especially short-term memory. Furthermore, the exergame modulates prefrontal brain activity during walking and enhances executive function in older adults [27,28].The rats stomping game: gait disorder is one of the most important risk factors for falls among the elderly. Studies have shown that elderly individuals with slow stride frequency and shorter stride length are more likely to fall. The goal of the rats stomping game was to improve gait among the elderly people. There were 9 holes in the ground; points were scored by stepping on rats that emerged from the holes. In the game, the participants had to raise their foot to a specified height and kill the rat by stomping on it 10 times. This strengthened their lower limb muscles and extended their stride. Visuospatial ability, attention, and executive function were also trained. Three exergame interfaces are shown in Figure 4.

#### 2.2.3. Training Feedback

During the feedback module, each score was recorded as a percentage and the participants could see their progress on a graph in real time. The participants could customize the training difficulty (easy, normal, and hard), switch virtual scenes, and background music, which increased the fun and motivation for training. The system also had self-adaptive capabilities. When the score was too low, it automatically reduced the difficulty of training.

## 3. Methods

According to ISO, user experience is a person’s perception and response to the use or anticipation of a product, system, or service, including users’ emotional performance, beliefs, preferences, cognition, physical/psychological reactions, behavior and achievements. There are various experience research frameworks in the market. They include System Usability Scale (SUS), Questionnaire for User Interaction Satisfaction (QUIS), The Standardized User Experience Percentile Rank Questionnaire (SUPR-Q), User Experience Questionnaire (UEQ), among others [29]. This research utilized the short version of the UEQ owing to its quickness and simplicity in capturing the user’s impression in experiencing the tools.

The User Experience Questionnaire (UEQ-S) [30] is a fast and reliable at measuring user experiences of interactive products, including pragmatic quality and hedonic quality. It is composed of eight items as shown in Table 2. The pragmatic quality aspect explains the technical focus of perception to achieve goals in a product, system, or service design. Likewise, the hedonistic quality aspect refers to non-technical aspects associated with the emotions of the user.

Several steps were utilized during the analysis of UEQ. The questionnaire answers were collected by assigning a value of 1 for the leftmost scale to a value of 7 for the rightmost scale option. The UEQ-S analytic tool [31], was used for data processing. The tool adjusted the score value in every statement answer from 7 Likert scales to −3 (most negative value) and +3 (most positive value). A simple heuristic was used to detect more or less random or not serious answers. The aim of detecting random or not serious answers was to check how much the best and worst evaluation of an item in a scale differed. If there was a big difference (>3), a problematic data pattern was anticipated. Inconsistent data were omitted from analysis. Values between −0.8 and 0.8 represented a neural evaluation of the corresponding scale, values >0.8 represent a positive evaluation, while values <−0.8 represented a negative evaluation. The scores between 1.5 and 2 indicated very good quality. The confidence interval and scale consistency were also computed. The confidence interval is a measure of the precision of the estimation of the mean. The smaller the confidence interval the higher the precision of the estimation and the more accurate the results are.

To verify whether the system was good or not, the UEQ-S value was compared with the benchmark data from [26]. This data set contains data from 14,056 persons from 280 studies concerning different products (business software, web pages, web shops, social networks). The comparison of the results for the evaluated product with the data in the benchmark allowed conclusions about the relative quality of the evaluated product compared to other products. An independent-sample t test was used to compare gender differences in user experience. The inclusion criteria for the participants were age (≥65 years) and prior experiences in playing exergames.

## 4. Results

Twenty-five participants (9 females, 16 males; mean ± SD age, 71.48 ± 4.09 years) were enrolled in this study. However, two participants were found to be inconsistent. Therefore, their data were omitted from the analysis. Data from 23 participants were used in the final analysis.

Table 3 shows the average scores of pragmatic quality, hedonic quality, and overall user experience (UX) scale at a 95% confidence interval. The scores indicate that the training system was given positive reviews by the users. The average score for each aspect were: pragmatic quality (1.652 ± 0.868), hedonic quality (1.880 ± 0.962), and the overall score was 1.776 ± 0.819. The overall score was higher than 0.8, which meant that the system gave a positive user experience.

The benchmark results from the UEQ Data Analysis Tool showed that the pragmatic quality score was good (10% of the results in the benchmark were better than the evaluated product, whereas 75% of the results were worse), while the score for hedonic quality was excellent (the evaluated product is among the best 10% of results). Comparatively, the impression regarding hedonic quality was distinctly higher than the impression regarding pragmatic quality (Figure 5).

An independent-sample *t*-test was used to compare gender differences regarding user experience. As shown in Table 4, t = 0.991, significance *p* = 0.333, the null hypothesis was invalidated, and there were no significant gender differences in pragmatic quality. As shown in Table 5, t = −0.092, significance *p* = 0.928, the null hypothesis was invalidated, and there were no significant gender differences in pragmatic hedonic quality.

## 5. Discussion

VR and AR are closely related, but with some differences. VR uses computer programmed interactive simulations to present opportunities for users to engage in environments that mimic real-world objects and events [32]. AR refers to the merging of a live view of the physical, real world with context-sensitive, computer-generated images to create a mixed reality. AR superimposes information to the real objects in real-time, while VR introduces the user to a fully synthetic world [33]. In terms of user experience, AR is more acceptable and suitable for the elderly compared to VR. Previous studies mainly adopted commercial VR exergame interventions that are mostly designed for entertainment purposes, rather than for the balanced rehabilitation of the elderly. Only a few exergames were designed specifically for the elderly [20,34]. Guimarães et al. [20], described an exergame for cognitive–motor training and fall prevention in older adults, including a diversity of tasks and interactive means. However, this exergame required the use of four wearable sensors placed on both arms and legs to support motion capturing and evaluation. This is invasive, and may make users uncomfortable. Marston et al. [34], developed an innovative Kinect-based iStopp-Falls system for continuous monitoring and reducing fall risk among elderly people. The system comprised four main components: training/physical tests, performance/feedback, learning/education, and meeting point. However, all the game scenes in the system were fully synthetic. Virtual interfaces are not very friendly for elderly people with less experience in games.

Therefore, we present an AR-based exergame system designed specifically for the elderly to reduce fall risk. The hardware is composed of a computer, an LCD, audio and Kinect 2.0 sensors. The software is made up of the following function modules: fall risk assessment, CMI training, and training feedback. In our design, we utilize Kinect 2.0, a low-cost, non-invasive, markerless camera, to capture and generate 3D models of the elderly by extracting them from the entire captured data and immersing them in interactive virtual environments. This approach is non-invasive because no external devices or sensors need to be worn by the elderly. Furthermore, using such live human models/avatars helps in understanding the emotional experiences of participants involved in the exergames [35]. In addition, the system highlights the synchronous interventions of cognition and motor. Traditional exercise therapy is mostly a single-factor intervention, while augmented reality therapy can carry out multi-factor interventions such as exercise and cognition. Some explicit cognitive training tasks for attention, memory, and executive functions are added to each game in the system.

The development of intelligent smart products for the elderly needs to meet functional requirements and to improve user experience according to their own characteristics and preferences. Measuring user experience is important before the software is fully implemented. UEQ provides an overview of the usability aspect to the user experience. The concept combines effectiveness and efficiency with aesthetic aspects, convenience of use and attractiveness. Effectiveness and efficiency are often referred to as pragmatic aspects, whereas aesthetics, convenience of use and attraction are referred to as hedonic aspects. According to the benchmark results through the UEQ Data Analysis Tool, it revealed a positive feedback from users. User experience between the genders was not significantly different. The impression regarding hedonic quality was significantly high compared to the impression regarding pragmatic quality. Improvements should be done in the future to enhance the prototype and increase its pragmatic quality. The design of these systems requires the participation of multidisciplinary research teams, including engineers, clinicians, cognitive psychologists, etc., in order to overcome the challenges associated with developing the system [36].

## 6. Conclusions

We developed an AR-based exergame system to reduce fall risk among the elderly from the perspective of user experience. Based on UEQ-S results, the user experience of the system is good enough. The major limitation of the study was the small sample size and smaller number of female users. Further investigations, using bigger sample sizes are recommended.

## Figures and Tables

**Figure 1 ijerph-17-07208-f001:**
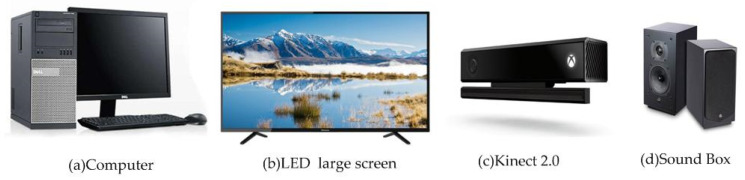
The hardware design of the system.

**Figure 2 ijerph-17-07208-f002:**
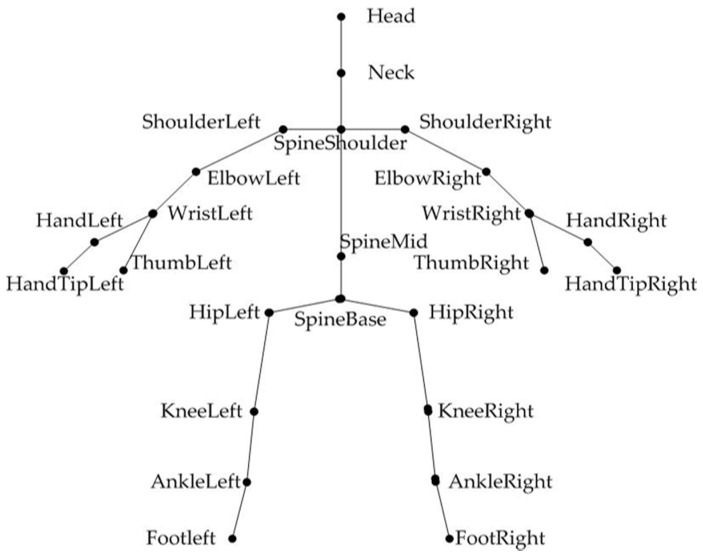
25-joint human skeleton model of Kinect2.0.

**Figure 3 ijerph-17-07208-f003:**
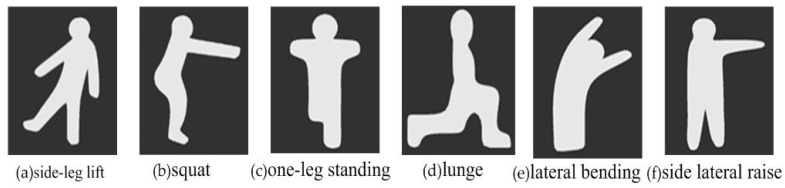
Shapes of the holes in the wall.

**Figure 4 ijerph-17-07208-f004:**
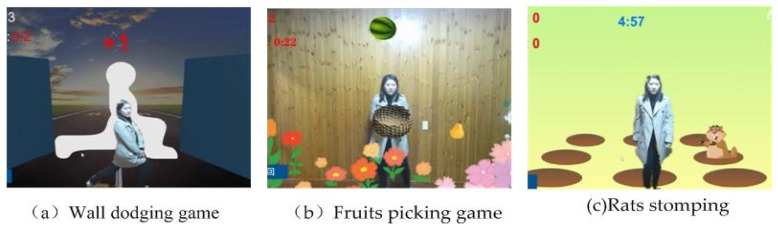
Three exergame interfaces.

**Figure 5 ijerph-17-07208-f005:**
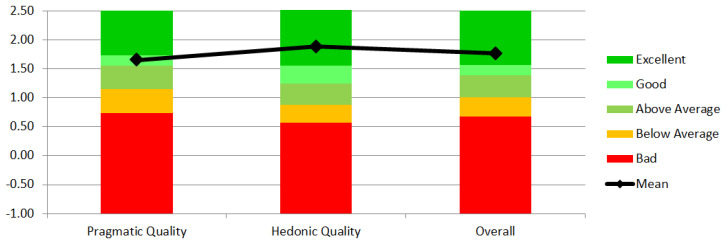
The benchmark result of the UEQ-S scales.

**Table 1 ijerph-17-07208-t001:** Exergames based on cognitive–motor intervention (CMI).

Exergames	Training Purposes		Game Description
	Motor Training	Cognitive Training	
Wall dodging	Muscle strength Balance Ability	Attention Executive function	The player must fit through the hole in each wall when it comes their way. Each wall presents a unique challenge. The participants have to think quickly and use motor dexterity to dodge each wall.
Fruits picking	Balance Ability Muscle strength	Short-term memory Attention Executive function	Three fruits are randomly displayed on the screen for five seconds; the participants should try to catch a specified fruit by moving the body from side to side.
Rats stomping	Gait Muscle strength	Visuospatial ability Attention Executive function	There are nine holes in the ground, and participants score points by stepping on rats that emerge from the holes, each rat has to be trampled 10 times.

**Table 2 ijerph-17-07208-t002:** UEQ-S evaluation items.

Item NO.	Evaluation									Quality Scale
	Negative	1	2	3	4	5	6	7	Positive	
1	obstructive	○	○	○	○	○	○	○	supportive	pragmatic
2	complicated	○	○	○	○	○	○	○	easy	pragmatic
3	inefficient	○	○	○	○	○	○	○	efficient	pragmatic
4	confusing	○	○	○	○	○	○	○	clear	pragmatic
5	boring	○	○	○	○	○	○	○	exciting	hedonic
6	Not interesting	○	○	○	○	○	○	○	interesting	hedonic
7	conventional	○	○	○	○	○	○	○	inventive	hedonic
8	usual	○	○	○	○	○	○	○	Leading-edge	hedonic

**Table 3 ijerph-17-07208-t003:** Mean for UEQ-S scale.

Confidence Intervals (*p* = 0.05) per Scale						
Scale	Mean	Std. Dev.	N	Confidence	Confidence Interval	
Pragmatic Quality	1.652	0.868	23	0.355	1.297	2.007
Hedonic Quality	1.880	0.962	23	0.393	1.487	2.274
Overall	1.766	0.819	23	0.335	1.432	2.101

**Table 4 ijerph-17-07208-t004:** *t*-test for pragmatic quality by gender.

	N	Mean	SD	t	*p*
Gender				0.991	0.333
Male	15	1.783	0.911		
Female	8	1.406	0.778		

**Table 5 ijerph-17-07208-t005:** *t*-test for hedonic quality–by gender.

	N	Mean	SD	t	*p*
Gender				−0.092	0.928
Male	15	1.867	1.035		
Female	8	1.906	0.876

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
