# Peer review of "Design and Evaluation of an Augmented Reality-Based Exergame System to Reduce Fall Risk in the Elderly"

_ijerph, 2020, doi:10.3390/ijerph17197208_

Round 1
Reviewer 1 Report
Dear authors,
Although you did a serious game to help people to be healthier, your paper has a lot of weaknesses.
Firstly, your game has nothing to do with virtual reality, it is not a totally immersive world, it can be a kind of augmented reality.
Secondly, your game helps people to be more healthy, and because they are healthier the have less possibilities to fall, not because of your system.
To sum up, you should change the purpose of your work because you did a serious game to help people to be active.
Author Response
Dear reviewer:
Thank you for your comments concerning our manuscript entitled “Design and evaluation of an augmented reality-based exergame system to reduce fall risk in the elderly” (ID: ijerph-918508). Those comments are all valuable and very helpful for revising and improving our paper, as well as the important guiding significance to our researches. We have studied comments carefully and have made correction which we hope meet with approval. Revised portion are marked in red in the paper.
Please see the attachment.
Once again, thank you very much for your comments and suggestions!
With best regards,
Yours sincerely,
Meiling Chen

Reviewer 2 Report
The paper describes an "exergame" (exercise game) that is meant the help to maintain/develop motor, balance and strength skills for elderly people under the risk of falling. The idea is good and the engineering work & user study in the paper are also good, however, scientifically the paper is not strong, as the novelty is not clear, comparison with related work is lacking, and the validation of the efficiency of the system is missing. I suggest the authors' continue their work in this area, and revise this paper to make it more publishable.
* There are quality scores given in the Abstract. It is not clean what the letter "M" with regard to those scores; it should be expanded.
* The paper should expand upon the differences with regard to related work, in particular, [22] and [23]. It it not sufficient just to mention these works, they should be compared and contrasted to the authors' exergame.
* The unique contributions of the authors' work should be stated explicitly in the introduction.
* The claims that the specific exercises improve health and other aspects e.g. "exercise also improves memory (especially short-term memory), attention, and executive function" should be substantiated, for example, by citations to existing literature or government / health institution guidelines.
* The effectiveness of the system is not assessed in the study, as admitted by the authors. If the effectiveness of the proposed exercises is better substantiated (see the previous point) it may not be a critical problem, however, the paper as it stands now offers not justification that the system actually works. This is not acceptable in my opinion, and should be improved upon, either by measuring the improvements after using the exergame, or by citing documents as suggested in the previous point.
Author Response

(The authors gave the same response as above.)

Round 2
Reviewer 1 Report
Dear authors,
You did a good job doing all the changes, but, this project doesn't give anything new to the state of art, you can find a lot of similar projects.
Kind regards
Reviewer 2 Report
The authors have engaged with all my comments. While so, I believe the scientific value of the study remains low.